# A Machine Learning Model Based on Thyroid US Radiomics to Discriminate Between Benign and Malignant Nodules

**DOI:** 10.3390/cancers16223775

**Published:** 2024-11-08

**Authors:** Antonino Guerrisi, Elena Seri, Vincenzo Dolcetti, Ludovica Miseo, Fulvia Elia, Gianmarco Lo Conte, Giovanni Del Gaudio, Patrizia Pacini, Angelo Barbato, Emanuele David, Vito Cantisani

**Affiliations:** 1Radiology and Diagnostic Imaging Unit, Department of Clinical and Dermatological Research, San Gallicano Dermatological Institute IRCCS, Via Elio Chianesi 53, 00144 Rome, Italy; antonino.guerrisi@ifo.it (A.G.); fulvia.elia@ifo.it (F.E.); 2Department of Radiological, Anatomo-Pathological Sciences, “Sapienza” University of Rome, Viale Regina Elena 324, 00161 Rome, Italy; seri.1901960@studenti.uniroma1.it (E.S.); vincenzodolcetti@gmail.com (V.D.); gianmarco.loconte@uniroma1.it (G.L.C.); g.d.gaudio@gmail.com (G.D.G.); patry.shepsut91@gmail.com (P.P.); emanuele.david@unict.it (E.D.); vito.cantisani@uniroma1.it (V.C.); 3Local Health Authority of Rieti, Via del Terminillo 42, 02100 Rieti, Italy; a.barbato@asl.rieti.it; 4Radiology Unit 1, Department of Medical Surgical Sciences and Advanced Technologies “GF Ingrassia”, University Hospital “Policlinico G. Rodolico”, University of Catania, 95123 Catania, Italy

**Keywords:** radiomics, ultrasound, machine learning, nodules

## Abstract

Thyroid nodules must be accurately classified as benign or malignant. The aim of this study is to develop a machine learning model based on thyroid ultrasound images in order to classify nodules into the two classes. Ultrasound (US) images were collected from 142 patients for training, validation and internal testing of three models, plus 21 images to externally test the best performing model. The random forest classifier model could perform the classification task, identifying all the malignant nodes and most of the benign.

## 1. Introduction

Thyroid nodules are a common clinical finding, with prevalence increasing with age and exposure to risk factors such as radiation. While most thyroid nodules are benign, a small percentage are malignant, necessitating accurate diagnostic methods to distinguish between benign and malignant lesions and guide appropriate clinical management [1].

The primary diagnostic tools for evaluating thyroid nodules include ultrasound (US), fine-needle aspiration (FNA) biopsy and cytological examination [2]. Ultrasound is the most widely used imaging modality used at the initial diagnostic phase for the nodule characterization due to its non-invasiveness, accessibility and ability to provide detailed information about the nodule’s size, composition and vascularity. However, ultrasound alone has limitations in accurately distinguishing benign from malignant nodules, often leading to many false positive nodules and, consequently, to unnecessary biopsies and surgeries.

FNA biopsy, often guided by ultrasound, is the gold standard for cytological evaluation of thyroid nodules [2]. Despite its high accuracy, FNA can yield indeterminate results in a significant number of cases, leading to diagnostic uncertainty and potential overtreatment. Thus, there is a pressing need for more precise, non-invasive diagnostic methods to improve the management of thyroid nodules [3].

In May 2014, the new classification method of thyroid cytology was published [4], which updated the previous classification with respect to literature data, making it comparable with the most used classifications (the American one, known as “Bethesda” [5,6], and the English one, of the Royal College of Pathologists of the United Kingdom (UKRCP) [7]) and providing a tool for endocrinologists and cytopathologists of immediate use for clinical practice. The scheme presents five categories, associated with the respective expected risk of malignancy and suggestion for clinical action (from TIR1 to TIR5). In this study, malignancy was associated with a TIR3B category and above, while benign nodules were identified as those with a TIR1-TIR3A stratification. As for imaging diagnostics, specifically ultrasound, there are many ultrasound parameters that can guide one’s steps towards a malignant or benign diagnosis, but they are not always univocal and there is no effective model that can distinguish between the two classes.

Radiomics could help in this endeavor, involving the extraction of a large number of quantitative features from medical images, capturing information about the texture, shape and intensity of the tissues that are not easily discernible and/or quantifiable qualitatively to the naked eye. These features can reflect the underlying pathophysiology of the tissues and have been shown to have potential in distinguishing benign from malignant lesions in various types of cancer [1].

Machine learning, particularly when combined with radiomic analysis, offers powerful tools to analyze complex and high-dimensional data. By training algorithms on labeled datasets, machine learning models can learn to recognize patterns associated with malignancy and predict the nature of new, unseen nodules with high accuracy [8].

This study aims to develop and validate a machine learning model based on radiomic features extracted from ultrasound images to classify thyroid nodules as benign or malignant, in order to avoid useless nodule biopsies. In ultrasound radiomics, image segmentation is used to obtain regions of interest (ROIs) for feature extraction. These ROIs may not necessarily be limited to thyroid nodules or tumors but can also encompass surrounding normal tissue. Information within these ROIs is leveraged to develop diagnostic, predictive or prognostic models [9]. The integration of radiomic analysis with machine learning has the potential to significantly enhance diagnostic accuracy, reduce the need for invasive procedures and improve patient outcomes [10]. In certain instances, in addition to improving diagnostic accuracy, a diverse array of radiomics features has even demonstrated the ability to correlate ultrasound data with anatomopathological data, allowing correlations with histological types of neoplasia with good precision; this is the case of the paper by Zhao et al., which evidenced the capability to effectively differentiate between medullary thyroid carcinoma (MTC) and papillary thyroid carcinoma (PTC) nodules of varying sizes [11]. By leveraging an ensemble of random forest classifiers, this study seeks to identify a robust and reliable model that can be applied in clinical settings.

Understanding the background and significance of this research underscores the potential impact of advanced diagnostic tools in thyroid nodule management, paving the way for more accurate, non-invasive and personalized patient care.

## 2. Materials and Methods

### 2.1. Patients and Image Acquisition

To train, validate and internally test the model, ultrasonography images were retrospectively collected from 142 Policlinico Umberto I patients that presented thyroid nodules. Among these subjects, 40 patients (28.2%) belonged to the “malignant” class (TIR 3B, 4 and 5) and 102 patients (71.8%) belonged to the “benign” one (TIR 1, 2 and 3A), according to histological diagnosis from fine-needle aspiration. This image set was used for the training, cross-validation and internal testing of 3 machine learning models based on different architectures. To externally test the model, an additional cohort of 21 patients from the same facility with suspicious thyroid nodules was enrolled in this study.

Only patients whose thyroid ultrasound images were free from annotation (clean from markers and needles) were chosen, to avoid fake density discontinuities.

### 2.2. Radiomic-Based Machine Learning Modelling

In agreement with the Image Biomarker Standardization Initiative (IBSI) guidelines [12], the radiomic methodology to collect, segment and analyze images was carried out by the IFO-San Gallicano institute, through the Trace4Research© radiomic software v. 1.0 component that allows one to obtain the whole IBSI-compliant radiomic workflow in a fully-automatic way and to generate the machine learning model for the thyroid nodule classification.

The IBSI-compliant radiomic workflow included the following: (i) a first step of segmentation of the Region Of Interest (ROI) for each ultrasound image, (ii) preprocessing of the segmented ROI (that is a requirement to compute features), (iii) feature extraction and selection and finally (iv) using those features to train, validate and internally and externally test different classifiers in the task of discriminating benign and malignant nodules in thyroid, thus comparing different machine learning models and choosing the best one on the basis of the best performance.

In this paper, The ROI segmentation was performed manually by two expert physicians in consensus (both with 15 years of experience), through the Trace4Research segmentation tool. The step of image intensities’ preprocessing within the segmented ROI included the resampling to isotropic pixel spacing, taking advantage of a down-sampling scheme built on the ROI greatest dimension (it caps the mask size at ten million pixels for the computation of texture features and at one million for other features). The Radiomics features, derived from the segmented ROI, belonged to seven different families: Intensity-based Statistics, Intensity Histogram, Gray-Level Co-occurrence Matrix (GLCM), Gray-Level Run Length Matrix (GLRLM), Gray-Level Size Zone Matrix (GLSZM), Neighborhood Gray Tone Difference Matrix (NGTDM), Neighboring Gray Level Dependence Matrix (NGLDM) and their definition, computation and nomenclature are compliant with the IBSI guidelines. It is worthwhile to specify that intensity histogram features were computed after an intensity discretization of the ROI, using a fixed number of 64 bins, that was also necessary for textural features computation (GLCM, GLRLM, GLSZM, NGTDM, NGLDM).

The features were then selected removing the ones that presented a low variance (<0.1) and high mutual information with the class label by a mutual information analysis (in which the mutual information threshold was 0.19). In this way, the resulting selected features are the most informative, non-redundant and reported according to IBSI standards.

Finally, three different models of machine learning classifiers were trained, validated and internally/externally tested, for the binary classification of malignant nodules vs. benign ones (based on supervised learning), using histological diagnosis from fine-needle aspiration as the reference standard. For each one of the models, a nested 4-fold cross validation method was used (two folds employed for training, one for validation and one for internal testing of each ensemble) and oversampling technique for the minority class (the malignant one) was applied through the adaptive synthetic sampling method (ADASYN).

Model 1 consisted of 4 ensembles of 16 random forest classifiers, combined with the Gini index with majority vote rule.

Model 2 was formed by 4 ensembles of 16 support vector machines combined with principal components analysis and the Fisher Discriminant Ratio with majority vote rule.

Model 3 consisted of 4 ensembles of 16 k-nearest neighbor classifiers combined with principal component analysis and the Fisher Discriminant Ratio with majority vote rule.

The performances of the 3 models were evaluated across all the folds in terms of majority vote and mean Area Under the Receiver Operating Characteristic Curve (ROC-AUC), accuracy, sensitivity, specificity, positive predictive value (PPV), negative predictive value (NPV) and their corresponding 95% confidence intervals (CIs).

The *p*-value for one-sided Wilcoxon signed rank test was measured to assess statistical significance, with respect to chance/random classification (null hypothesis of 50% for ROC-AUC, accuracy, sensitivity and specificity; the percentage of the benign subjects over the whole dataset −72% for PPV; and the percentage of the malignant subjects over the whole dataset −28% for NPV).

The model with the best performance, according to the ROC-AUC index, was chosen as the best classification model for the binary classification in malignant or benign nodules.

The best model was then externally tested with an additional cohort of 21 patients with thyroid nodules.

### 2.3. Statistical Analysis

Statistical analysis was conducted with embedded tools of the Trace4Research platform for the training, validation and internal testing, while the external testing evaluation was performed using SPSS software, v. 26. To describe the distribution of each of the most relevant features in the “malignant” and “benign” classes, their medians with 95% CI were calculated and presented using graphical violin and box plots for easier visualization and interpretation. A non-parametric univariate Wilcoxon rank-sum test (Mann–Whitney U test) was performed for each of the relevant radiomic predictors to verify its significance in discriminating the two “malignant” and “benign” classes. To take into account multiple comparisons, the *p*-values were adjusted using the Bonferroni–Holm method and the significance levels were set at 0.05 (*) and 0.005 (**).

## 3. Results

### 3.1. Radiomic-Based Machine Learning Modelling

From each segmented ROI of every image involved in this study, 777 IBSI-compliant radiomic features were extracted. For the classification task of interest (102 images from the class “benign” vs. 40 images from the class “malignant”), these features were used to train, cross-validate and internally test (nested 4-fold cross validation) the three aforementioned different models of machine learning classifiers.

Table 1 1-2-3 show the values for ROC-AUC, accuracy, sensitivity (identifying the benign class), specificity (identifying malignant class), positive predictive value (PPV) and negative predictive value (NPV) as obtained from the training, cross-validation and internal testing of models 1, 2 and 3, which all consisted of four ensembles of machine learning classifiers. Furthermore, for each model, ROC curves for the four ensembles are plotted in Figure 1A–C.

Based on ROC-AUC values for internal testing, the model formed by random forest classifiers (model 1) turned out to be the best model to discriminate malignant from benign nodules.

External testing was also performed on the first and best classification model, consisting of 4 ensembles of 16 random forest classifiers. Images from a cohort of 21 patients were used for this task (29% in the malignant and 71% in the benign class), and the performance in terms of accuracy, sensitivity, specificity, PPV and NPV metrics is described in Table 2.

### 3.2. Radiomic Predictors

The 11 selected radiomic predictors are listed in Table 3, complete with their IBSI feature family and feature nomenclature. Those predictors are ranked according to their statistical significance, and to their frequencies among the most relevant ones in the ensemble of model 1, the one constituted by random forest classifiers. Median values, 95% CIs [13] and results from univariate statistical sum rank tests are also reported with adjusted *p*-values for each feature. The violin plot and boxplot of the radiomic predictors are shown in Figure 2.

The eleven listed features were all very significative, with a corrected *p*-value lower than 0.005. In Figure 2, violin and box plots for these features and for the first model are shown, differentiated for the two classes.

Model 1 performed very accurately in both internal and external testing, and the 11 features that it used as predictors were all correlated with the classification task at hand.

## 4. Discussion

In this paper, the possibility of classifying benign and malignant thyroid nodules from ultrasound imaging by radiomic analysis combined with machine learning is evaluated. Radiomic analysis, being a feature-based tool capable of detecting sometimes impossible to see differences and nuances and discerning repetitive and/or characteristic patterns and measures in the region of interest, limits interobserver variability by providing an objective metric for the classification task [14], and, given there is sometimes tumor heterogeneity, it helps address biopsy results that come from only one tissue sample [15].

In the realm of thyroid cancer diagnosis via ultrasonography, several recent papers have explored the utility and accuracy of radiomics and machine learning approaches, which provides a foundation for placing our findings in a broader context. Zhao and Ma’s study [11] compared the radiomics features between different sizes of medullary thyroid carcinoma (MTC) and papillary thyroid carcinoma (PTC) tumors. Their analysis yielded significant insights into the differentiating features for these tumor types, emphasizing that a combination of features outperformed individual features in discriminating between MTC and PTC macronodules and micronodules. This highlighted the increased diagnostic accuracy when combining multiple radiomic features, akin to our ensemble model’s superior performance over single classifier models.

Chang et al. (2016) [15] developed a computer-aided diagnosis (CAD) system using ultrasound images to differentiate between benign and malignant thyroid nodules. Their system utilized various imaging features and a support vector machine (SVM) classifier, achieving a maximum accuracy of 98.3% with leave-one-out cross-validation. These findings underline the high potential of CAD systems to match or even surpass radiologist accuracy, as our best ensemble model also demonstrated (ROC-AUC of 85). Our approach aligns with their achievements by employing machine learning to enhance diagnostic accuracy.

Lambin et al. [16] provide a broader perspective on radiomics as a bridge to personalized medicine, emphasizing the need for standardized evaluation in radiomics investigations. Their paper stresses the importance of rigorous validation which resonates with our methodological approach, ensuring our ensemble models were not only optimized but subjected to rigorous cross-validation and an external test cohort.

Liang et al. [17] compared their developed radiomics score against the 2017 ACR TI-RADS scoring criteria, showcasing a superior performance of their radiomics-derived model. Our ensemble approach similarly demonstrated that machine learning models, specifically ensembles of classifiers, can outperform traditional scoring systems like TI-RADS evaluated by less experienced radiologists, reinforcing the potential of automated systems in improving diagnostic precision.

Zhou et al. [10] proposed deep learning radiomics (DLRT) models for differentiating benign from malignant thyroid nodules. Their results, showing high AUC values across training, internal and external validation cohorts, corroborate the efficacy of advanced deep learning models over traditional assessment methods. Our study, albeit using classical machine learning techniques rather than deep learning, reflects similar trends in achieving high diagnostic performance, highlighting the versatility and effectiveness of these approaches.

Finally, Gild et al. [18] focused on the risk stratification of indeterminate thyroid nodules using ultrasound and machine learning algorithms. Their use of both TI-RADS and deep learning models resulted in promising predictive values, particularly for certain patient subsets. Our study’s consistent performance metrics in both internal and external validation support the viability of machine learning models in refining preoperative diagnostics and reducing unnecessary surgeries.

In conclusion, our findings are corroborated by the literature’s progressive understanding that combining advanced radiomic features with machine learning techniques can significantly enhance the accuracy of thyroid nodule classification. These studies collectively underscore the pivotal role of machine learning and radiomics in modernizing and improving the diagnostic process for thyroid cancer.

In the present study, the performances of three different machine learning classifiers are evaluated that employed radiomic features extracted from ultrasound images of thyroid nodules. First, the three models were trained, validated and internally tested on 142 nodule images, divided into malignant (28.2%) and benign (71.8%) thanks to information brought by fine-needle aspiration. The three models were composed of four ensembles of random forest, support vector machine and k nearest neighbor classifiers, respectively.

The best performing model turned out to be the first one, and metrics were evaluated for each stage of the classification process: in training, it achieved 100% for all metrics, while in validation the performances gave an 84% ROC-AUC value, an 81% accuracy value, a 67% sensitivity value, an 84% specificity value, a 63% PPV and an 87% NPV. The first testing of the models was internal, and the top performer is again the one formed by four ensembles of sixteen random forests, with metrics expressed both in mean values (that were always significative with a *p*-value < 0.005) and in majority vote (that corresponded to a 50% threshold), achieving good results, in line with the ones obtained in the aforementioned studies.

This promising model was finally tested on an external cohort of 21 patients, suffering from one nodule each, belonging to both malignant and benign classes. External testing evaluation was performed, resulting in 90.5% accuracy, 100% sensitivity, 86.7% specificity, 75% PPV and 100% NPV: these metrics values confirm the high performances of the random forest model in discriminating the two nodule classes investigated in this paper, correctly classifying all of the malignant nodules and the majority of benign.

The eleven features which were most relevant in the classification task were all significant to the univariate statistical sum rank analysis, reporting a Bonferroni-corrected *p*-value inferior to 0.005. Three of these radiomic predictors belong to the Neighboring Grey Level Dependence Matrix family that identifies how many times the same value occurs in neighboring pixels, two to the Neighborhood Grey Tone Difference Matrix family, three to the Grey-Level Co-Occurrence family, one to the Grey-Level Run Length Matrix family, one to the Intensity Histogram family and one to the Intensity Based Statistics family. The violin box plots shown in Figure 2 represent how the cases are categorized for each radiomic feature, and their statistical distribution: in this way, the difference in values for the two classes is made immediately noticeable.

The most statistically relevant features are denominated “US gradient dependence count energy” and “US square complexity”: in the former, the gradient filter enhances sudden intensity changes and the energy gives information on the image pattern uniformity: a high value indicates that the contrast variation in the images is distributed in a more homogeneous way and with similar variations. In the latter, the square filter enhances the image contrast, and the complexity is a feature again related to the sudden variability in intensity values. In this specific case, the values indicate that the malignant nodules are characterized by a more random pattern in terms of contrast variation (hypo- and hyper-echogenic areas do not have a repetitive scheme), presenting a lower value that indicates that tumor heterogeneity plays an important role in discerning between the two classes. In the third to ninth feature, the Laplacian of Gaussian filter is implicated, which makes “edge detection” possible: from the data, it seems that malignant nodules have bigger homogeneous areas associated with fewer sudden changes in intensity, and that they present a higher count of small and concentrated hypo-echogenic areas with high contrast. Summing up, confirmation is found that, in a more homogeneous and echogenic background, malignant nodules present a higher variability in terms of pixel intensity distributions due to many small darker areas.

By comparing the results obtained from the models with the evaluations carried out by the operator who performed the ultrasound scans using the ACR TI-RADS classification system (an expert operator with more than twenty years of experience in the field), placed in turn to compare with the result of the cyto-histological examination, good agreement was observed (diagnostic accuracy 90%, sensitivity 88%, specificity 92%, PPV 86%, NPV 93%).

However, these values tend to deviate when the evaluation is carried out by operators with less experience but also using different TI-RADS classification systems, which use different evaluation criteria and above all are affected by the subjective evaluation of the operator (such as echogenicity and appearance of the margins). The combination of these factors tends to increase the degree of inter-operator variability, complicating the diagnostic process and the possible surgical evaluation of the patient, aspects on which ML models can have a positive impact by using qualitative and quantitative features for the evaluation of the nodules that are more objective and reproducible [19].

There are some limitations to this study: first, ultrasound is an imaging technique which strongly depends on the operator’s expertise; second, the segmentation of the lesion was performed manually and could be influenced by the same specific experience of the operator who acquired the images: however, this problem is partially addressed by the Trace4Research software v. 1.0 that automatically performs slightly different re-segmentations in the image pre-processing phase to mimic the work of distinct physicians, thus selecting robust features for the model. In the opinion of the authors, this limitation is, however, secondary compared to the evaluation of standard parameters that are used in the clinical setting. Furthermore, given the small cohort available for this study, additional external tests should be performed, with images from different ultrasound machines and different healthcare centers to increase the variability of the target population.

## 5. Conclusions

The goal of this work was to develop a machine learning model to classify malignant and benign thyroid nodules only from ultrasound imaging. Three different models were trained, and the best model was found to be the one constituted of four ensembles of random forest classifiers that, through 11 radiomic features with high statistical significance, could successfully identify all the malignant nodes and the consistent majority of benign in the external testing cohort.

Being able to determine malignant from benign nodules without the need for fine-needle biopsy would shorten diagnosis timing and thus bring forward therapy initiation, while being a totally non-invasive procedure. Further investigations could be conducted by testing the model with images of nodules from different centers.

## Figures and Tables

**Figure 1 cancers-16-03775-f001:**
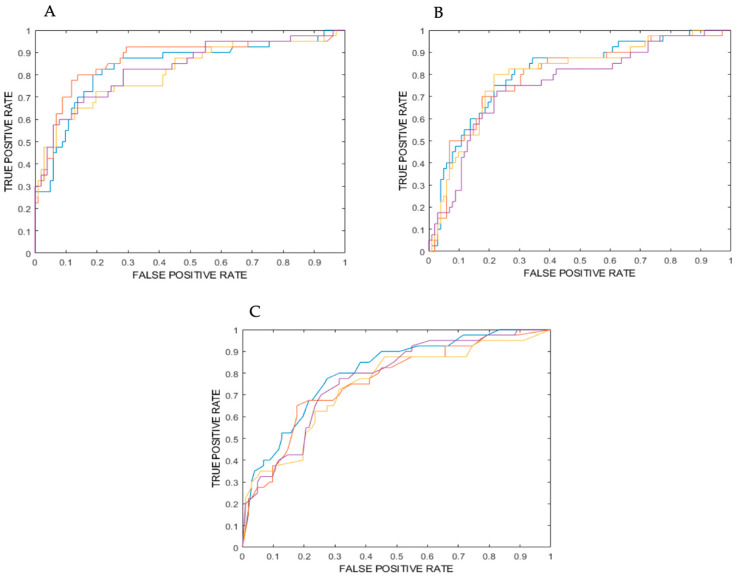
(**A**) ROC curve for the model consisting of 4 ensembles of random forest classifiers (from internal testing). (**B**) ROC curve for the model consisting of 4 ensembles of support vector machine classifiers (from internal testing). (**C**) ROC Curve for the model consisting of 4 ensembles of K nearest neighbors’ classifiers (from internal testing). Purple: ensemble 1; yellow: ensemble 2; blue: ensemble 3; orange: ensemble 4.

**Figure 2 cancers-16-03775-f002:**
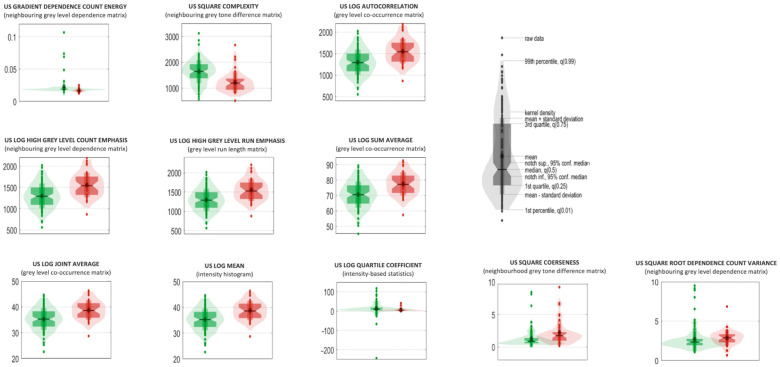
Violin and box plots of the radiomic predictors for model 1 (ensemble of random forest; violin and box plots of “benign” and “malignant” classes are reported in red and green, respectively).

**Table 1 cancers-16-03775-t001:** Model 1, 2 and 3 performances. Classification performance in terms of AUC, accuracy, sensitivity, specificity, PPV, NPV, corresponding 95% confidence interval and statistical significance with respect to chance/random classification (*p*-value). Performances are reported for training, validation and internal testing.

Model	Metric	Training	Validation	Internal Testing (Mean)	Internal Testing (Majority Vote)
1	ROC-AUC (%) (95% CI)	100 ^1^ [99,100]	83 ^2^ [81–84]	84 ^2^ [80–87]	85
	Accuracy (%) (95% CI)	100 ^1^ [99,100]	79 ^2^ [79,80]	81 ^2^ [77–85]	83
	Sensitivity (%) (95% CI)	100 ^1^ [99,100]	67 ^2^ [81–84]	68 ^2^ [64–71]	70
	Specificity (%) (95% CI)	100 ^1^ [99,100]	84 ^2^ [83–85]	87 ^2^ [82–91]	88
	PPV (%) (95% CI)	100 ^1^ [99,100]	63 ^2^ [60–66]	66 ^2^ [58–75]	70
	NPV (%) (95% CI)	100 ^1^ [99,100]	87 ^2^ [86–89]	87 ^2^ [86–89]	88
2	ROC-AUC (%) (95% CI)	84 ^2^ [83–86]	81 ^2^ [79–83]	79 ^2^ [76–83]	80
	Accuracy (%) (95% CI)	77 ^2^ [76–78]	73 ^2^ [71–76]	75 ^2^ [72–77]	75
	Sensitivity (%) (95% CI)	78 ^2^ [78,79]	74 ^2^ [72–77]	76 ^2^ [69–83]	78
	Specificity (%) (95% CI)	76 ^2^ [74–78]	73 ^2^ [70–76]	74 ^2^ [71–77]	75
	PPV (%) (95% CI)	77 ^2^ [76–78]	53 ^2^ [49–57]	54 ^2^ [51–56]	54
	NPV (%) (95% CI)	77 ^2^ [77,78]	88 ^2^ [87–89]	89 ^2^ [86–92]	89
3	ROC-AUC (%) (95% CI)	95 ^2^ [94–96]	62 ^2^ [60–64]	77 ^2^ [73–81]	79
	Accuracy (%) (95% CI)	87 ^2^ [85–88]	58 ^2^ [56–61]	71 ^2^ [67–74]	75
	Sensitivity (%) (95% CI)	96 ^2^ [95–98]	57 ^2^ [47–68]	71 ^2^ [60–82]	73
	Specificity (%) (95% CI)	77 ^2^ [74–79]	59 ^2^ [53–64]	71 ^2^ [66–76]	75
	PPV (%) (95% CI)	81 ^2^ [79–83]	36 ^2^ [35–37]	49 ^2^ [45–53]	54
	NPV (%) (95% CI)	96 ^2^ [94–97]	79 ^2^ [75–82]	86 ^2^ [82–90]	88

^1^ *p*-value < 0.05/^2^ *p*-value < 0.005.

**Table 2 cancers-16-03775-t002:** Model 1 external testing evaluation.

Metric	External Testing of Model 1
Accuracy (%) (95% CI)	90 [77–100]
Sensitivity (%) (95% CI)	87 [73–100]
Specificity (%) (95% CI)	100 [100]
PPV (%) (95% CI)	100 [100]
NPV (%) (95% CI)	75 [54–96]

**Table 3 cancers-16-03775-t003:** Ensemble of random forest classifiers: the 11 predictors sorted in descending order according to their statistical significance and relevance.

#	Feature Family	Feature Nomenclature	Median in the Benign Class (95% CI)	Median in the Malignant Class (95% CI)	Uncorrected *p*-Value	Corrected *p*-Value
1	Neighbouring Grey Level Dependence Matrix	US_gradient_dependence Count Energy	1.63 × 10^−2^ [1.56 × 10^−2^–1.69 × 10^−2^]	1.87 × 10^−2^ [1.83 × 10^−2^–1.91 × 10^−2^]	<0.005	<0.005
2	Neighbourhood Grey Tone Difference Matrix	US_square_complexity	1189.3 [1082.33–1296.26]	1641.68 [1552.62–1730.75]	<0.005	<0.005
3	Grey-Level Co-Occurrence Matrix	US_LoG_autocorrelation	1542.46 [1433.46–1651.45]	1289.97 [1224.34–1355.59]	<0.005	<0.005
4	Neighbouring Grey Level Dependence Matrix	US_LoG_high Grey Level Count Emphasis	1544.4 [1437.37–1651.44]	1292.93 [1227.6–1358.26]	<0.005	<0.005
5	Grey-Level Run Length Matrix	US_LoG_High Grey Level Run Emphasis	1525.45 [1419.45–1631.44]	1289.15 [1225–1353.3]	<0.005	<0.005
6	Grey-Level Co-Occurrence Matrix	US_LoG_sum Average	77.19 [74.38–79.99]	70.51 [68.64–72.38]	<0.005	<0.005
7	Grey-Level Co-Occurrence Matrix	US_LoG_joint Average	38.59 [37.19–40]	35.25 [34.32–36.19]	<0.005	<0.005
8	Intensity Histogram	US_LoG_mean	38.58 [37.18–39.98]	35.24 [34.31–36.17]	<0.005	<0.005
9	Intensity-Based Statistics	US_LoG_Quartile Coefficient	4.48 [3.47–5.48]	9.96 [7.75–12.16]	<0.005	<0.005
10	Neighbourhood Grey Tone Difference Matrix	US_square_coarseness	1.73 × 10^−3^ [1.40 × 10^−3^–2.06 × 10^−3^]	8.66 × 10^−4^ [7.35 × 10^−4^–9.96 × 10^−4^]	<0.005	<0.005
11	Neighbouring Grey Level Dependence Matrix	US_squareroot_dependence Count Variance	2.88 [2.62–3.14]	2.32 [2.19–2.45]	<0.005	<0.05

## Data Availability

The raw data supporting the conclusions of this article will be made available by the authors on request.

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
