# Peer review of "A Machine Learning Model Based on Thyroid US Radiomics to Discriminate Between Benign and Malignant Nodules"

_cancers, 2024, doi:10.3390/cancers16223775_

Round 1

Reviewer 1 Report

Comments and Suggestions for Authors

This is an important article that tests different machine-learning algorithms to descriminate beween benign and malignant thyroid nodules. Although I agree with the findings and think that this article has major merits,
I would like to raise some questions to the authors.

1. It is not clear to me how the training and validation data sets were created.  How many patients of each group were included in the training and how many in the validation step ?
2. Internal testing:  can the authors explain what internal testing means ?
3.  The legends of table 1A,B,C mention that the p-values reflect significance with respect to chance/random classification.  Which percentage of accuracy, sensitivity, specificity, PPV or NPV do the authors assume in their null hypothesis ?
Naively, I would, for example, use a PPV of 50% as the PPV obtained under random testing.  I see in Table 1C that the PPV for the internal validation has a confidence interval of [45,53].  This means that the PPV is not significantly different from 50%, but the p-value is, according to the table highly significant.
4. Please provide a legend for the 4 different colors in Figures 1A-C.
5. Table 2: Please provide confidence intervals
6. The model was tested on 40 patients with malignant nodules and 102 patients with benign modules.  It was externally validated on a data set of 21 patient with both benign and malignant nodules. The test population is clearly different from the validation population.  Was there a reason for this and if so, can the authors explain ?  If there is no reason, can the authors describe if the external cross-validation would be different if tested in a population of patients with only benign and patient with only malignant nodules ?
7. Tabel 3:  the 95% CI was calculated for each of the 11 predictors.  How was this confidence interval calculated ?

In addition, I have some orthographic suggestions:
7. Section 2.1, first sentence:  To train, validate and internal test the model, ...: write 'internally', not 'internal'
8. Section 3.1, third paragraph, first sentence:  ... for internal testing, ,  the model...: leave one comma out

Reviewer 2 Report

Comments and Suggestions for Authors

The aim of this study was to develop a machine learning model able, basing on thyroid ultrasound images, to characterize nodules into benign and malignant classes. In this respect, quantitative feature extraction from medical images (US Radiomics) was applied to an ensemble of 142 subjects whose nodules resulted to be malignant (28.2 %) or benign (71.8%) according to ultrasonographic images and histological diagnosis from fine-needle aspiration. In such context, the used radiomic approach was thought to may discriminate the disease heterogeneity between the two groups. The above image set was employed for the training, cross-validation and internal testing of three different machine learning models, which consisted of 4 ensembles of machine-learning classifiers in order to develop the binary classification task of interest. The best performing model was then externally tested on a cohort of 21 new patients. It was concluded that the model with 4 ensembles of random forest classifiers was able to identify all the malignant nodules and the majority of the benign ones in the external testing cohort.

Introduction deals with diagnostic criteria, classification methods and clinical course of thyroid nodules evidencing how radiomic analysis, carried out on ultrasound images and combined to machine learning, is likely to improve diagnostic accuracy, decrease invasive procedures, ameliorate patient outcomes and also correlate with anatomopathological data.

Materials and Methods report patients and image acquisition, radiomic-based machine-learning modelling (RBMLM) and statistical analysis.

Results extensively describe RBMLM with the support of five tables (table 1A/B/C and tables 2 and 3) and tour figures (figure 1A/B/C and figure 2).

Discussion analyses the obtained results also considering literature data dealing with radiomics and machine learning approaches in thyroid cancer diagnosis. Authors underline how they evaluated three different machine-learning classifiers by employing radiomic features extracted from ultrasound images of thyroid nodules, thus concluding that the best model was the one constituted of 4 ensembles of random forest classifiers considering 11 radiomic features with high statistical significance.

On the whole, this research is interesting and well performed on a methodological and technical point of view. The manuscript has been prepared with accuracy and language clearness. There are no particular remarks as well as lexicon, “English style”, sentence fluency, tables and figures and their legends, and references are concerned.

Reviewer 3 Report

Comments and Suggestions for Authors

The main objective of the research is to develop a machine learning model based on thyroid ultrasound images to classify nodules into benign or malignant classes. Three models of machine-learning classifiers (random forests, support vector machines and k-nearest neighbor classifiers) were developed for the binary classification task of interest.

The data was of high quality and clearly illustrated. The conclusions are consistent with the evidence and arguments presented. The references are appropriate. Overall, this work is suitable for publication after addressing a few minor concerns.

Comments and suggestions:

1.      When the abbreviations are first mentioned, the full name should be introduced, for example ‘US images’ in the Simple Summary, ‘ROC-AUC’, ‘PPV’, and ‘NPV’ in the Abstract.

2.      Table 1 can be combined in one table with one more column with three classifiers.

3.      Put Figure 1A-C in one figure.

4.      For Figure 2, the titles of the first three figures are partially cropped, also with some numbers obscured in the other two figures. I have included screenshots of them below. The last gray violin and box plots are also introduced, showing detailed views of the violin and box plots.

A screenshot of a graph

Description automatically generated

5.      Drew a brief conclusion and discussion after summarizing the results.

6.      A thyroid nodule can be diagnosed using a combination of fine-needle aspiration biopsy (FNA) imaging techniques such as ultrasonography, CT, MRI and PET–CT. The patients’ sample in this study may also be tested with other methods to finally diagnose thyroid nodule. Is it possible to combine the diagnosed results from US with other method for more accuracy classification?

In summary, I recommended minor revisions to this article before its publication in Cancers.

Reviewer 4 Report

Comments and Suggestions for Authors

The study addresses a significant need in thyroid nodule diagnostics by incorporating ML and radiomics, methods that could outperform traditional scoring systems such as TI-RADS.

I would like to comment on the following points.

Please clarify whether the patients evaluated in the internal and external tests of this study were recruited from a single facility. 

Given the limited number of patients assessed in the external test, it may be necessary to conduct multi-center studies in the future to generalize these findings. Including this as a limitation in your discussion is recommended.

Additionally, in your conclusion, you note that this technique improves thyroid nodule classification accuracy. However, please also clarify how this knowledge may benefit future patient management.
